# CodeBPE: Investigating Subtokenization Options for Large Language Model Pretraining on Source Code

**Nadezhda Chirkova**
Naver Labs Europe[*]
nadia.chirkova@naverlabs.com

**Sergey Troshin**
University of Amsterdam[*]
s.troshin@uva.nl

## Abstract

Recent works have widely adopted large language model pretraining for source code, suggested source code-specific pretraining objectives and investigated the applicability of various Transformer-based language model architectures for source code. This work investigates another important aspect of such models, namely the effect of different subtokenization options, and aims at identifying most effective and length-efficient subtokenizations, taking into account code specifics. We propose subtokenziation that reduces average length by 17% without downstream performance drop, and show that a carefully chosen subtokenization may improve quality by 0.5-2%, possibly with some length increase.

## 1 Introduction

With the inspiration from the success of large language model (LM) pretraining in natural language processing (NLP), BERT-like models have been widely adopted for source code processing (Feng et al., 2020; Kanade et al., 2020), as code has a similar discrete sequential structure to natural text. Being trained on huge source code corpora in a self-supervised manner, large LMs often substantially outperform domain-specific models developed purposely for applied tasks, especially in the tasks with limited parallel / labelled data (Ahmad et al., 2021a). These tasks include fixing code bugs, generating text from code and vice versa, or translating code between programming languages.

Recent works advanced large LM pretraining on source code in two main directions. First, various model kinds were utilized for source code: CodeBERT (Feng et al., 2020) and CuBERT (Kanade et al., 2020) rely on the classic encoder-only RoBERTa (Liu et al., 2019), CodeGPT (Lu et al., 2021) uses decoder-only GPT (Radford & Narasimhan, 2018), PLBART (Ahmad et al., 2021a) is based on the denoising sequence-to-sequence BART (Lewis et al., 2020) model, and CodeT5 (Wang et al., 2021b) utilizes multitask sequence-to-sequence T5 (Raffel et al., 2020). Second, a range of code-specific self-supervised pretraining tasks were proposed to enrich the classic masked language modeling (MLM) objective, e. g. GraphCodeBERT (Guo et al., 2021) predicts data flow connections during pretraining (one variable is computed from another variable), and CodeT5 (Wang et al., 2021b) and DOBF (Roziere et al., 2021) use a variable naming objective.

This work is devoted to investigating one more important component, subtokenization, which is usually not paid much attention when pretraining large LMs on source code. Modern LMs usually preprocess sequences using open-vocabulary models such as Byte-pair encoding (BPE, Sennrich et al., 2016) which split long tokens into smaller subtokens. Though this process is often referred to as tokenization, we call it subtokenization, to underline its smaller granularity. Subtokenization became a standard part of all widely-used LMs pretrained on natural text or code, because it ensures the relatively high frequency of all subtokens (compared to the whitespace-separated tokenization, which results in a large portion of out-of-vocabulary tokens), at the same time producing sequences of reasonable length (compared to character-level tokenization). Though subtokenization was initially introduced for NLP, it is especially relevant for code, as programming languages usually permit identifiers of unrestricted complexity, e. g. variable or function names (Chirkova & Troshin, 2021).

---

[*]Work done while being at HSE University.

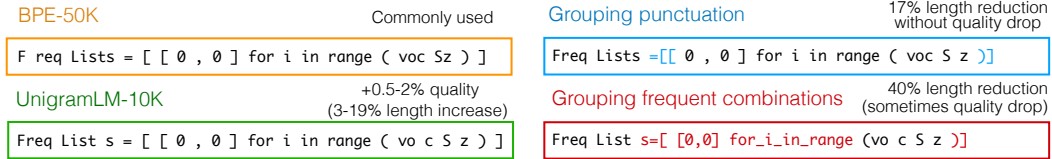

Figure 1: Example subtokenizations (all numbers compared to the commonly used BPE-50K).

Though subtokenization is often chosen with only superficial deliberation, it is one of the essential model components which may affect both quality and prediction speed. First, an inaccurately chosen subtokenization procedure may substantially increase sequence lengths and consequently slow down prediction. As a simple example, the work on CodeT5 (Wang et al., 2021b) notices that using BPE trained specifically on source code corpora makes sequences 30–45% shorter than using BPE trained on natural text. Second, a line of recent research points at the positive effect of the carefully chosen subtokenization procedure on the model's performance in NLP. For example, Bostrom & Durrett (2020) show that using a UnigramLM (Kudo, 2018) subtokenization algorithm instead of BPE improves the quality of BERT-based question answering or textual entailment in English by 1%, and Ding et al. (2019) show that adjusting BPE vocabulary size in translation may produce +4 BLEU. At the same time, for large LMs, the particular subtokenization procedure chosen at the pretraining stage becomes an inseparable part of the model and must later be used in applied tasks. This underlines the need for a careful choice of subtokenization options when pretraining large LMs.

In this work, we conduct a deep study of subtokenization options for large LM pretraining on source code, using PLBART as a testing ground. In addition to investigating general aspects, e. g. the subtokenization algorithm and the vocabulary size, we study the ways of adapting subtokenization to the specific properties of code, such as a large amount of punctuation marks and frequently-used token combinations, a variety of complex identifiers, or relative similarity of programming languages. We aim at choosing optimal subtokenization options that (a) lead to the best performance or (b) minimize sequence lengths (and thus speed up the model) without downstream performance drop. Our contributions are as follows - we show that for large LMs pretrained on source code:

- Grouping punctuation chars in single tokens reduces the average length by 17% without downstream performance drop (we call this approach CodeBPE or CodeUnigramLM), and permitting more complex composite tokens reduces lengths by 40%, sometimes with quality drop (Section 1);
- UnigramLM is generally preferable over BPE (Section 4);
- Smaller vocabularies may improve quality with 3–19% length increase (Section 5);
- Subtokenizers are well transferable between programming languages (Section 6);

Our length-efficient subtokenization procedure (see examples in Figure 1) compresses sequences by 17% without quality drop and our most effective subtokenization improves performance by 0.5–2% significantly in three out of eight tasks and by one standard deviation in two other tasks.

## 2 METHODOLOGY AND EXPERIMENTAL SETUP

The existing works on large LMs for source code usually choose a particular subtokenization library, for example the same as in the base LM the work uses, and train the subtokenizer with the vocabulary size of 30-50K on source code corpora used for pretraining. Often code is pre-processed before subtokenization, e. g. by replacing `\n` with `NEW_LINE`, and split into tokens on white-spaces and punctuation marks so that these tokens are further split into subtokens, e. g. `for i in range (vocSize)` will be split into [`for`, `i`, `in`, `range`, `(`, `vocSize`, `)`] even if `for i in` is generally a frequent combination. The latter principle appears to be intuitively reasonable, since it ensures that subtokenization preserves syntactically meaningful boundaries of tokens (Kanade et al., 2020). We refer to this principle as prohibiting *composite tokens*. More details on subtokenization in different LMs for code are given in Section 7.

We treat the described commonly-used approach as a baseline, and conduct a series of experiments, each modifying the baseline subtokenization procedure in one dimension and pretraining PLBART with the new subtokenization. The dimensions we vary are as follows: the allowed complexity of composite tokens, the subtokenization algorithm, the vocabulary size, the set of languages the subtokenizer is trained on, and the use of stochastic subtokenization. These dimensions are inspired either by the specifics of source code or by the recent works on subtokenization in NLP.

**Experimental setup.** As our base model, we use PLBART (Ahmad et al., 2021a), since it comes with the released pretraining code and data preprocessing routine under the MIT license. We use the same model size, the pretraining dataset size and other hyperparameter settings, including finetuning hyperparameters, as in PLBART[1]. In particular, we use an encoder-decoder Transformer architecture with 6 layers in each part, with the model dimension of 768 and 12 heads (140M parameters). The pretraining data consists of 230M Python functions, 470M Java functions (crawled through BigQuery[2]) and 47M natural language (NL) descriptions (crawled from StackOverflow[3]), referred to as sequences below. The BigQuery dataset consists of repositories with clear open-source license. We pretrain all our PLBART models for 100k updates, as in the original paper.

As applied tasks, we consider three tasks from the PLBART paper: code generation (generating a Java function based on an NL description; CONCODE (Iyer et al., 2018) dataset, CodeBLEU (Ren et al., 2020) metric), code summarization (generating an NL description for a Python or Java function; CodeSearchNet (Husain et al., 2020) dataset, BLEU metric), code clone detection (classifying whether two Java functions implement the same functionality; BigCloneBench dataset (Svajlenko & Roy, 2015); F1 metric), and one additional task of code translation (translating code from Python to Java and vice versa; AVATAR dataset (Ahmad et al., 2021b)). Here we consider original data with the CodeBLEU metric (Code Translation-1) and the smaller version of data with tests and the Computational Accuracy metric – which portion of generated functions passed all tests (Code Translation-2). We chose tasks so that we have both code generative and discriminative tasks and that datasets are either in Python or Java.

We clip all sequences by 510 subtokens, except summarization where we clip by 250 subtokens following Ahmad et al. (2021a). Such clipping remains the majority of sequences unclipped in all subtokenizations: 96-99.1% in the pretraining data, 93–99% in translation, 88–100% in generation, 76–93% in summarization, and 37–80% in clone detection. In the main text we report average lengths computed on the randomly chosen subset of pretraining data before clipping, Appendix A reports length statistics for downstream data with similar trends as observed for the pretraining data. We only clip sequences passed to neural networks and use unclipped target sequences when computing metrics.

**Baseline subtokenization.** Following Ahmad et al. (2021a), we use a SentencePiece (Kudo & Richardson, 2018) library, which is a one of the most widely used solutions for subtokenization. We train subtokenizers on 10M functions and NL descriptions randomly selected from the pretraining data (different from the random subset on which we measure average lengths). Though Ahmad et al. (2021a) use BPE subtokenization algorithm, our baseline subtokenization uses another algorithm, UnigramLM, because it was shown to be quantitatively and qualitatively more suitable for pretraining in NLP than BPE (Bostrom & Durrett, 2020). We also perform their comparison for code in Section 4. We set the vocabulary size to 50K (the commonly used size for large LMs of code) and character coverage to 99.99% (enough to cover English chars and punctuation).

We also use PLBART's preprocessing which includes removing comments and docstrings, replacing \n, indents and dedents in Python with NEW_LINE, INDENT and DEDENT tokens as they are a part of the language syntax, and removing formatting in Java as it does not affect the language syntax. Our baseline subtokenizer follows the commonly used strategy of prohibiting composite tokens described above. The only exception we make is that we allow underscores _ inside tokens, because they do not represent a syntax unit, as other punctuation chars do.

---

[1] https://github.com/wasiahmad/PLBART
[2] https://console.cloud.google.com/marketplace/details/github/github-repos
[3] https://archive.org/download/stackexchange

Table 1: Different levels of allowed composite tokens complexity considered in the paper. **Green** emphasizes tokens which could not be obtained in the previous level, and **gray** emphasises the remaining tokens that could not be obtained in Level 0. Levels list *allowed* merges, but what particular merges to perform is chosen by the tokenizer.

| Lev. | Description | Example |
|---|---|---|
| 0 | Whitespaces in the middle of tokens are prohibited and each punctuation char is treated as a separate token (except '_') | ['for','i','in','range','(','df','.','shape','[','1',']',')', ':','NEW_LINE','INDENT','print','(','i',')','NEW_LINE', 'print','(','df','.','columns','[','i',']',')'] |
| 1 | Similar to Level 0, but tokens consisting of several punctuation chars are allowed | ['for', 'i', 'in', 'range', '(', 'df', '.', 'shape', '[', '1', '**]）:**', '**NEW‧LINE INDENT**', 'print', '(', 'i', '**)‧NEW‧LINE**', 'print', '(', 'df', '.', 'columns', '[', 'i', '**])**'] |
| 2 | Similar to Level 1, but dots are allowed in tokens | ['for', 'i', 'in', 'range', '(', 'df', '**.shape**', '[', '1', '**] ) :**', '**NEW‧LINE  INDENT**', 'print', '(', 'i', '**) NEW‧LINE**', 'print', '(', 'df', '**.columns**', '[', 'i', '**])**'] |
| 3 | Whitespaces and single punctuation chars allowed in tokens, except NEW_LINE | ['**for i in range**', '**( df**', '**. shape [ 1**', '**] ) :**', '**NEW‧LINE INDENT**', 'print', '**( i**', '**) NEW‧LINE**', 'print', '**( df**', '**. column**', '**s [ i**', '**] )**'] |
| 4 | Composite tokens of arbitrary complexity are allowed | ['**for i in range**', '**( df**', '**. shape**', '**[ 1 ]**', '**)**', '**: NEW‧LINE**', '**INDENT print**', '**( i )**', '**NEW‧LINE print**', '**( df**', '**. columns**', '**[ i ] )**'] |

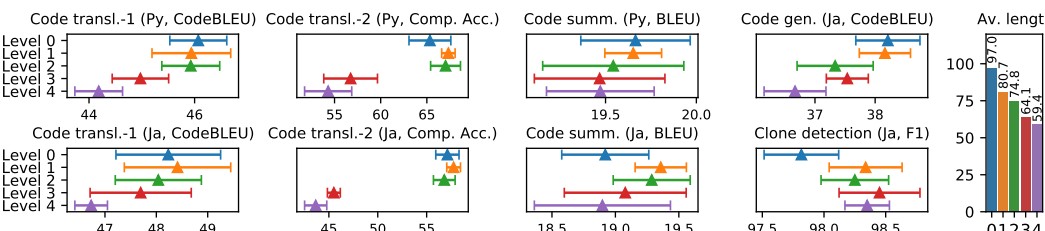

Figure 2: Results on various subtokenization granularity, averaged over 4 finetuning runs (mean $\pm$ standard deviation). Level 0 – baseline subtokenization. Numerical data for all plots is given in Appendix.

# 3 SUBTOKENIZATION GRANULARITY

In natural text, a portion of punctuation chars is small and thus their separation in subtokenization does not affect lengths much. In contrast, in source code, punctuation constitutes 12.8% of chars and often forms frequent combinations joining which into composite tokens may substantially reduce lengths. Further, the presence of a large amount of commonly used patterns is another specific feature of source code, e. g. `for (int i = 0;` in Java or `def __init__ (self):` in Python, and these patterns again may form composite tokens. This section investigates the effects of the use of composite tokens on performance and length-efficiency.

We consider several levels of allowed complexity of composite tokens listed in Table 1 and empirically compare them in Figure 2. The two extreme cases are no composite tokens (Level 0, equal to the baseline subtokenization) and unrestricted composite tokens complexity (Level 4, composite tokens constitute 48.6% of the vocabulary). The average sequence length in Level 4 is 40% less than that in Level 0. At the same time, the effect on performance depends on the task: in code-generative tasks (translation and generation), Level 4 performs significantly worse than Level 0, and in code understanding tasks, Level 4 is either similar / marginally worse than Level 0 (code summarization) or even significantly better (clone detection). Because of quality loss encountered in several tasks, we consider intermediate levels.

Level 1 makes one step further from Level 0 and allows punctuation char merges, e. g. '`})`' or '`]):`'. Though such punctuation composite tokens constitute only 3.4% of the vocabulary, their use reduces average length by 17%: from 97 to 80.7, and since this level does not mix punctuation with other chars, it presumably should not complicate code generation much. Level 2 makes one more

step further and allows merging dots `.` with textual tokens. This reduces the average length by 23% compared to Level 0. The motivation for Level 2 is that a lot of API name tokens almost always go with the dot, e. g. `.join` or `.split` in Python. Figure 2 shows that Level 1 model performs similar or better than Level 0 model in all tasks, and Level 2 performs similar or better than Level 0 in six tasks, marginally worse in Python code summarization and significantly worse in Java code generation.

Level 3 makes a step back from Level 4 and restricts the complexity of composite tokens in such a manner that each composed token may represent either a simple one-line code pattern or a punctuation combination, but could not combine them. Quantitatively, Level 3 performs generally better than Level 4, but (marginally or significantly) worse than the previous Level 2 in six tasks and similarly in two tasks (generation and clone detection).

To sum up, *punctuation combinations (Level 1) result in sequence lengths reduction by 17% without performance drop in all tasks. We verify this result for BPE in Appendix B and call this approach CodeBPE or CodeUnigramLM. Length reduction could be increased up to 24% in most tasks by allowing dots attached to tokens (Level 2) and up to 40% in most code understanding tasks by allowing arbitrary subtoken combinations (Level 4).* we investigate the transferability of subtokenizers between programming languages in Section 6.

One of the potential issues with using composite tokens in code-generative tasks is that an inaccurate generation of a "long" token may change the entire following generated code. For example, in Java–Python code translation, a cycle which traverses all unique element pairs in an array, converts to

```
for l in range ( 0 , arr_size - 1 ) :
  for r in range ( l + 1 , arr_size ) :
```

While the Level 0 model generates exactly the specified cycle and the Level 1 model only modifies the first cycle: `range ( arr_size - 1 )`, making it even more concise, Level 3 model generates

```
for l in range ( 0 , arr_size ) :
  for r in range ( 0 , arr_size ) :
```

which results in traversing some elements twice. Here the first cycle begun with tokens 'for l in' and 'range ( 0 ,' and the second cycle begun with tokens 'for r in' and 'range ( 0 ,' where the latter repeats the previously used token and starts an incorrect line. However, according to our manual prediction analyses, such inaccurate generation, if it happens, rarely results in wrong code and often does not affect code semantics. For example, the Level 3 model may generate ['range ( 0 ,', 'n )'] instead of equivalent `range(n)`. Another example is that this model may generate `[ [ 0 ] * c for i in range ( r ) ]` instead of two nested cycles by beginning with tokens '[ [' and '0 ] *', resulting in even more concise code.

As for composite tokens in Level 1, they contain only punctuation and are "simpler" than in Level 3. Besides, Level 1 composite tokens serve more often for statement closing (e. g. ') ) :' at the end of the cycle specification) than for a harder starting of new statements: 46.3% of Level 1 composite tokens contain only closing brackets, 12.8% contain only opening brackets and 26.7% contain both. We also check that using punctuation composite tokens does not deteriorate syntactic correctness: in Java-Python code translation-1, Level 0 and Level 1 models generate a similar number of syntactically correct test code snippets: 1226 and 1239 correspondingly. At the same time, for the Level 3 model, this quantity only equals 1163.

In Appendix C, we also analyse how much do input and output subtoken sequences intersect in different Levels and find that generally the higher granularity leads to the lower intersection rate. This may be another explanation for the superiority of the lower granularity subtokenizations, as intuitively it should be easier for the model to predict correct subtokens if they are present in the input sequence.

As aurogregressive decoding is a slowest part of the encoder-decoder pipeline (Berard et al., 2021), it is important to check that the length statistics of sequences *generated* by the models comprising composite tokens are close to those of the data. We check it for Java-Python translation-1: while groundtruth sequences at Levels 1 and 3 are 13.5% and 50% shorter than at Level 0, the generated sequences at these levels are 15% and 40% shorter than sequences generated at Level 0.

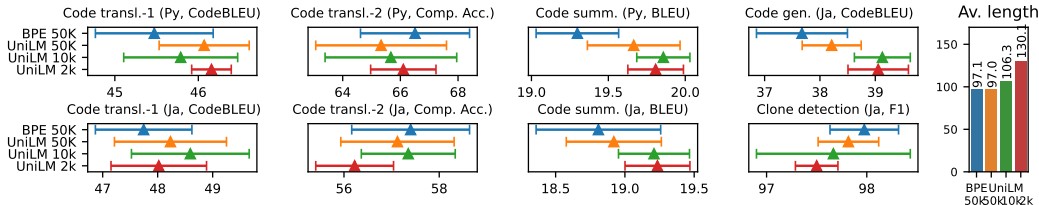

Figure 3: Comparison of BPE and UnigramLM subtokenizers and of several vocabulary sizes. UnigramLM 50K – baseline subtokenization.

Table 2: Example subtokenization of identifiers by UnigramLM and BPE subtokenizers

| Original token | UnigramLM subtokenization | BPE subtokenization | Native subtokenization (Camel- or snake_case) |
|---|---|---|---|
| `fromDottedString` | ['from', 'Dotted', 'String'] | ['from', 'Dot', 'ted', 'String'] | ['from','Dotted','String'] |
| `isInstantiated` | ['is','Instantiate','d'] | ['isIn','stanti','ated'] | ['is','Instantiated'] |
| `GridBagConverter` | ['Grid', 'Bag', 'Converter'] | ['GridBag','Converter'] | ['Grid','Bag','Converter'] |
| `isSameSize Horizontally` | ['isSame', 'Size', 'Horizontally'] | ['isSame', 'Size', 'H', 'orizontally'] | ['is', 'Same', 'Size', 'Horizontally'] |
| `PA_Hierarchy_ID` | ['PA','_','Hierarchy','_ID'] | ['PA', '_H', 'ierarchy', '_ID'] | ['PA', '_', 'Hierarchy', '_', 'ID'] |

We note that though we use sequence clipping which clips lower granularity subtokenizations stronger, it does not devalue our results, as generally lower granularity subtokenizations perform better in our experiments, and also such clipping is a widely used practical scenario; we provide more comments in Appendix C. At the same time, in rare cases when lower granularity subtokenizations perform worse than higher granularity ones, this may be indeed due to the max-length clipping. For example, we find that the more "fair" cropping eliminates the superiority of Level 1 compared to Level 0 in code summarization and they start performing similarly (see Appendix C).

## 4 Subtokenization algorithm

Bostrom & Durrett (2020) compare two most popular subtokenization approaches, BPE and UnigramLM (Kudo, 2018), for pretraining of large LMs on natural text data. While BPE constructs the vocabulary in the bottom-up fashion, starting from characters and gradually joining them, the UnigramLM algorithm works in the top-down fashion, staring from a large vocabulary and gradually filtering it. The paper finds that UnigramLM outperforms BPE in a range of downstream tasks and suggests several reasons for the superiority of UnigramLM, including better alignment with morphology and the more efficient vocabulary allocation. Since most existing pretrained LMs on source code use BPE, we decided to compare the two algorithms for source code.

Figure 3 compares BPE and UnigramLM for PLBART. In three tasks, UnigramLM outperforms BPE by one standard deviation, and in remaining tasks their performance is very close. Since the average length of two tokenizations is similar, *we recommend using UnigramLM for source code*.

Bostrom & Durrett (2020) argue that one of the potential reasons for the superiority of UnigramLM subtokenization is that it is better aligned with natural text morphology and thus simplifies the composition of words by parts. We find that a similar effect appears for identifiers in source code: although 80% of identifiers are subtokenized identically by UnigramLM and BPE, for some of the remaining 20%, UnigramLM provides more "reasonable" splits into subtokens, see examples in Table 2. More formally, we observe that UnigramLM subtokenization better resembles splitting into subtokens based on `CamelCase` or `snake_case`, which we call a native subtokenization. To estimate this effect quantitatively, we consider the Python corpus and randomly select a set of 150K identifiers with different UnigramLM and BPE subtokenizations consisting of $\geqslant 2$ native subto-

kens, and measure the average Jaccard similarity $J(A, B) = |A \cap B|/|A \cup B|$ between the set of native subtokens and the set of subtokens produced by each subtokenizer. The resulting score for UnigramLM, 26.6%, is much higher than for BPE, 15.2%. As could be observed from the third and the fourth rows in Table 2, sometimes subtokenizers join two native subtokens into one (`isSame`, `GridBag`). If we split each subtoken produced by a tokenizer based on `CamelCase` or `snake_case` to eliminate this effect and again measure average Jaccard similarities, UnigramLM's score, 55.2%, is still much higher than BPE's, 47.9%, again indicating that UnigramLM's tokenization is better aligned with the native one. In Appendix C we measure intersections between inputs and outputs in the sequnce-to-sequence tasks and find that UnigramLM leads to a slightly higher intersection rate than BPE, which may be connected to the better alignment with native subtokenization and serve as a a possible explanation of UnigramLM slight performance superiority.

A relatively frequent pattern is that BPE tends to detach the first uppercase letter from native subtokens (`H orizontally` in row 4, `_H ierarchy` in row 5). Among 150K identifiers considered in the previous paragraph, 14.6% of BPE tokenizations contain at least one single uppercase letter `X` and 4.4% — at least one subtoken of kind `_X`, while for UnigramLM these scores are substantially lower and equal to 11.8% and 1.4% correspondingly. At the same time, BPE merges two native subtokens more frequently (`GridBag` in row 3): 45.8% BPE tokenizations contain at least one token which could be split into two or more based on `CamelCase`, while for UnigramLM this score only equals to 39.2%.

## 5  VOCABULARY SIZE

This section studies the effect of vocabulary size, one of the main subtokenizer's hyperparameters, on the downstream quality of PLBART. Though the existing pretrained LMs for code use relatively large vocabularies of 30–50K tokens, we are interested, whether using smaller and less length-efficient vocabularies could result in better performance, and if yes, how large is the length increase.

Figure 3 presents the comparison of PLBARTs trained with vocabulary sizes 50K (large), 10K (medium) and 2K (small). We find that in code translation, all vocabularies lead to similar performance. In code summarization, small and medium vocabularies outperform the large one by one standard deviation. In code generation, the medium vocabulary significantly outperforms the large one. Finally, in clone detection, decreasing the vocabulary size deteriorates quality. At the same time, with the large vocabulary, sequences are shorter than with the smaller vocabulary by 9.5% (10K) and 33% (2K), but the model size is larger (139M for 50K, 108M for 10K, and 102M for 2K). We conclude that *vocabulary size reduction may lead to a slight performance improvement but with sequences elongation*, thus it may be helpful in applications with high cost of errors and weak restrictions on sequences lengths. We verify the highlighted result for BPE in Appendix. We note that compared to BPE 50K which is used in most existing large LMs of code, UnigramLM 10K improves performance significantly in three tasks and by one standard deviation in two other tasks.

Reducing vocabulary size increases the granularity of identifiers subtokenization, e. g. `reachable` is subtokenized as `reachable` with the 50K vocabulary, `reach able` – with 10K and `re ach able` – with 2K. In other words, vocabulary size reduction may be seen as an even stronger prohibition of complex tokens than Level 0 in Section 1. Our results on the effectiveness of smaller granularity agree with the machine translation results of Ding et al. (2019). Programs in code generation and summarization data are more identifier-centered, e. g. the model often needs to choose a correct API based on the natural language description which seems to be easier by composing from smaller subtokens. On the contrary, in code translation, data is more algorithmic-centered, with mostly short identifiers encoded in 1–2 subtokens with all vocabulary sizes. The length increase of 10K vocabulary compared to 50K one is 6–19% in the former two tasks (6% in generation, 19% in summarization) and only 3.5% in the latter one (code-translation-1).

## 6  TRANSFERABILITY BETWEEN PROGRAMMING LANGUAGES

Due to the high computational cost of large LM pretraining and relative programming languages similarity, e. g. compared to how dissimilar natural languages could be, pretrained LMs on source code are often used for programming languages that were not considered during pretraining. In this

Figure 5: Results of transferability between programming languages. Py+Ja – subtokenizer is trained on all data (baseline), Only Py – on Python and natural language data only.

section, we investigate the effect of using a subtokenizer trained on one programming language for another programming language.

Figure 4 visualizes the number of tokens having particular frequencies in Python and Java languages, and black rectangles denote language-specific areas. We find that the baseline Level 0 granularity vocabulary seems to be language-universal: the majority of subtokens have large frequencies in both languages, and only a small number of subtokens, 12.6%, are frequent in one language and rare in another. Interestingly, for Level 4 vocabulary, this quantity is not much higher, 20.1%, though it should include all language-specific composite tokens. As composite tokens occupy almost half of the Level 4 vocabulary, the remaining 30% composite tokens are common for two languages.

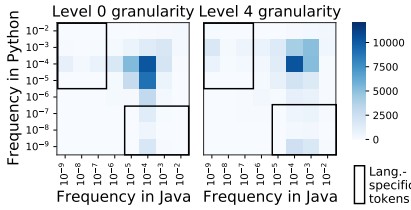

Figure 4: Number of tokens of different frequency in two languages, UnigramLM 50K vocabularies.

Analysing sequence lengths (Figure 5), we observe that training the subtokenizer without Java (*Only Py*) shortens Python sequences marginally and increases Java sequences by 6.5% compared to the baseline subtokenizer trained on all data (*Py+Ja*). The latter happens because some widely used Java identifiers were not merged into single tokens as they are not used in Python; still, the length increase is not so large. For the Level 4 granularity subtokenizer, *Only Py*'s length increase on Java is larger, 13%, since it contains more language-specific composite tokens. However, due to common composite tokens, the resulting Level 4 *Only Py*'s Java average length is still smaller than Level 1 *Only Py*'s Java sequences: 79 vs. 83.

As for performance, using the *Only Py* subtokenizer instead of *Py+Ja* changes quality up to one standard deviation and could both increase and decrease it on Java data (quality increase may be caused by the increased subtokenization granularity). Note that we only change subtokenizer configuration – PLBART is still pretrained on all languages, this may happen in practice if LM's developers use the subtokenizer from another project, e. g. for comparison purposes. Summing up, we conclude that *the baseline subtokenizer is universal and, if needed, could be used for other programming languages it was not trained on, with small length increase and slight quality change*. We note that Python and Java have quite different syntax and are usually used in different applications.

## 7 RELATED WORK

**Subtokenization studies for NLP.** Subtokenization has become an essential component of modern NLP pipelines and thus — a subject of a line of empirical NLP studies. While word-based models suffer from the out-of-vocabulary problem, subtoken-based (open-vocabulary) as well as char-based approaches cover arbitrary novel words. Among various open-vocabulary approaches, BPE (Sennrich et al., 2016), WordPiece (Wu et al., 2016) and UnigramLM (Bostrom & Durrett, 2020) became most widely used, and UnigramLM was shown to outperform BPE for LM pretraining (Bostrom & Durrett, 2020). A line of studies investigate the optimal granularity of word subtokenization: Ding et al. (2019) find that in Transformer-based neural machine translation, small vocabularies of 0–4K subtokens outperform large ones by up to 4 BLEU, and VOLT (Radford et al., 2018) automates the search of a proper subtoken vocabulary with a proper size by formulating it

as an optimal transport problem. The smallest char-based granularity is often avoided because of substantial sequences elongation, but has particular strengths, e. g. much less number of hyperparameters and better robustness, and thus appears to be a promising research direction (Gupta et al., 2019; Clark et al., 2021; Tay et al., 2021). Provilkov et al. (2020); Kudo (2018) propose stochastic subtokenization as a way to improve new words composition and (Wang et al., 2021a) adapt it to pretrained LMs. Finally, an actively studied challenge is that various natural languages need different subtokenization decisions and are hard to subtokenize with one common model (Chung et al., 2020; Rust et al., 2021). Our work investigates most of the specified directions for source code. For a more detailed review on subtokenization, see (Mielke et al., 2021).

**Subtokenization practices in neural source code processing.** Subtokenization was first tested for source code in (Karampatsis et al., 2020) and later used in the majority of Transformer-based models. Almost all LMs pretrained on source code use BPE-like subtokenization with large vocabulary: CodeBERT uses the WordPiece (Wu et al., 2016) algorithm (a modified BPE, 50K), CuBERT – an algorithm from the Tensor2Tensor project (Vaswani et al., 2018) (50K), PLBART and CodeGPT – BPE (50K), CodeT5 – byte-level BPE (32K), DOBF uses a subtokenization procedure of either CodeBERT or Roziere et al. (2020) (BPE 64K) for fair comparison, AlphaCode Li et al. (2022) – SentencePiece (8K, algorithm not specified), InCoder Fried et al. (2022) – BPE (50K). To the best of our knowledge, existing works do not provide an in-depth experimental analysis of various subtokenization options for code and, particularly, do not investigate various levels of composite tokens complexity. Though the concurrent work of Fried et al. (2022) uses unrestricted composite tokens (our Level 4), they do not compare them to any other subtokenizations. Level 4 composite tokens are conceptually similar to code idioms used in (Iyer et al., 2019; Shin et al., 2019) for code generation, but the mentioned works develop specific procedures for mining idioms, which need separate implementation, while we rely on the commonly-used subtokenization procedure.

## 8 CONCLUSION

In this work, we conducted an empirical study of varying subtokenization options for large LMs pretraining on source code. We believe that main the value of our work is not in improved numerical criteria, but importantly in providing *reference experiments* for the community showing which impact (both substantial or small) subtokenization choices may have in pretraining LMs for code. This underlines which directions to look more carefully at in practice (the use of composite tokens) and which are less important to experiment with (vocabulary size, subtokenization algorithm), and whether the later directions can bring at least a slight improvement or not (yes, they can). Currently most works experiment with subtokenizations options only superficially or do not experiment at all, and we hope that our work will provide motivation to do that.

**Our recommendations.** As for particular direct recommendations from our results, first, we recommend to use the proposed punctuation combination approach, which we call CodeBPE or Code-UnigramLM depending on the used subtokenization algorithm, that shortens sequences by 17% without quality drop. We suppose that with larger pretrained LMs higher levels of composite tokens may also achieve comparable performance; we were not able to experiment with then as they require very extensive computational resources. Second, if changing the subtokenization algorithm is easy, e.g. when using the SentencePiece library, we recommend using the UnigramLM, since it performs slightly better than commonly used BPE with similar lengths. Third, we recommend considering releasing models with smaller vocabularies, as they may perform slightly better than larger vocabularies. In our experiments the UnigramLM-10K subtokenizer was 0.5–2% more effective than the commonly-used BPE 50K in 5/8 experiments, but with 3.5–19% length increase.

**Limitations** The main work's limitation is that we consider only the PLBART model, due to the limited computational resources. However, we believe that the provided recommendations will motivate and simplify the process of the subtokenizer's tuning for future works, as described above. Another limitation is that we focus on finding optimal subtokenization options only for source code, though some downstream tasks also include the processing of natural language. Investigating the ways of choosing optimal subtokenization for both code and natural language may be an interesting direction for future research. Finally, we only compare BPE and UnigramLM, while it could be interesting to investigate the performance of other algorithms, e. g. WordPiece.

## BROADER IMPACT

We do not anticipate any direct negative social impact of our work. However, our results may potentially be used for developing new pretrained LMs for source code, and a detailed discussion on their broader impact is provided in Chen et al. (2021) (Section 7), e. g. over-reliance on generated code or producing vulnerable code. Unfortunately, our work may cause negative environmental impact because of computation ($\sim$5K Tesla A-100 GPU hours and $\sim$4K Tesla V-100 GPU hours at the internal cluster).

## ACKNOWLEDGMENTS

The results were supported by the Russian Science Foundation grant №19-71-30020. The research was supported in part through the computational resources of HPC facilities at NRU HSE.

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

## A   AVERAGE LENGTHS ON THE FINETUNING DATA

In the main text, we report average lengths computed over a randomly chosen subset of the pre-training data (different from the random subsets subtokenizers were pretrained on). Table 3 reports average lengths computed over finetuning data, we also include average lengths computed over held-out pretraining data. For programming languages, we observe similar trends as on the general pretraining data: Level 1 compresses sequences by 11.6–17.7% (the lower compression in the code summarization task, 11.6–12.8%, is explained by the higher percent of identifiers in this data than in other tasks); Level 4 compresses sequences by 32.4–50%; 10K and 2K vocabularies increase lengths by 3.5–10.2% and 12.7–35.9% respectively; and BPE and UnigramLM average lengths are similar.

As for natural text, its average lengths are not affected by the Level 1 subtokenization because it only affects punctuation char sequences, rarely present in the natural language data. At the same time, higher level subtokenizers may produce natural language subtoken sequences of higher lengths, because their vocabularies are occupied by programming-focused composite tokens rarely present in natural language data, while frequent words may be absent in these vocabularies. Small vocabulary subtokenizers also have programming language-focused vocabularies, resulting in higher lengths increase for natural text than for programming languages.

In Table 4, we report time needed to generate predictions for the test set in two tasks. The measurement was conducted on a single Tesla V100 GPU, in the same session for all runs. The speed-up is stronger than the length decrease, because of quadratic Transformer complexity.

## B   ADDITIONAL EXPERIMENTS WITH BPE

Figure 6 presents the comparison of BPE Level 0 and Level 1 subtokenizations and of BPE 50K and 10K vocabularies. The results are similar to those of UnigramLM reported in the main text: the performance of Level 0 and Level 1 subtokenizations is again close, the model with 10K vocabulary again significantly outperforms the model with 50K vocabulary in code generation and outperforms by one standard deviation in Java code summarization, and in other tasks the performance of 10K and 50K models is similar.

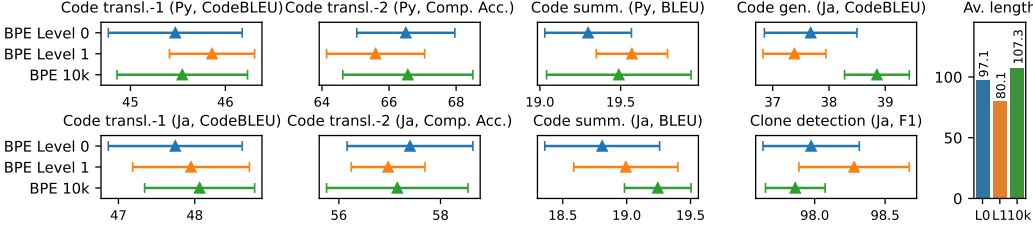

Figure 6: Comparison of BPE Level 0, BPE Level 1 (both with 50K vocabulary as in the main text) and BPE 10K.

## C   ADDITIONAL ANALYSIS

**Additional experiments with "fair" subtoken sequence cropping.**    In our main experiments, we clip all subtoken sequences with the same maximum length threshold, as this scenario is most close to how it is done in practice. It should be noted that for the vast majority of examples, subtoken sequences fit to the maximum length limit and are not affected by this clipping (see statistics in the main text, Section 2). However, for the remaining long examples, subtoken sequences produced by subtokenizers of higher granularity may be detokenized into longer character sequences, than that of the smaller granularity subtokenizer's. For example, if one uses the maximum length of 4 (only used for illustration), then the code snippet `x = sum ( numbers )` may be tokenized by the Level 4 subtokenizer into ['x =', 'sum (', 'numbers', ')'] (fits into the limit) and by 2K subtokenizer into ['x', '=', 'sum', '(', 'numbers', ')'], which will be clipped into ['x', '=', 'sum', '('] with

Table 3: Average lengths computed over finetuning data in different tasks. All subtokenizations expect the last one use the UnigramLM algorithm; Base ans Level 1–4 subtokenizations use vocabulary of 50K; 10K and 2K subtokenizations are based on Level 1 preprocessing..

| Subtok. | Base | Level 1 | Level 2 | Level 3 | Level 4 | 10K | 2K | BPE |
|---|---|---|---|---|---|---|---|---|
| Code translation – Python | | | | | | | | |
| Av. len | 150.6 | 126.9 | 125.1 | 91.5 | 87.1 | 156.0 | 169.7 | 150.5 |
| vs. Base | 0.0% | -15.7% | -16.9% | -39.2% | -42.2% | +3.6% | +12.7% | -0.1% |
| Code translation – Java | | | | | | | | |
| Av. len | 226.8 | 193.9 | 173.6 | 127.3 | 113.5 | 234.7 | 263.0 | 226.8 |
| vs. Base | 0.0% | -14.5% | -23.5% | -43.9% | -50.0% | +3.5% | +16.0% | 0.0% |
| Code summarization – Python | | | | | | | | |
| Av. len | 151.4 | 133.9 | 125.7 | 107.0 | 102.3 | 166.3 | 202.1 | 151.3 |
| vs. Base | 0.0% | -11.6% | -17.0% | -29.3% | -32.4% | +9.8% | +33.5% | -0.1% |
| Code summarization – Java | | | | | | | | |
| Av. len | 132.7 | 115.7 | 109.4 | 95.0 | 84.9 | 146.3 | 180.4 | 132.5 |
| vs. Base | 0.0% | -12.8% | -17.6% | -28.4% | -36.0% | +10.2% | +35.9% | -0.2% |
| Code summarization – natural text | | | | | | | | |
| Av. len | 12.5 | 12.5 | 13.3 | 17.8 | 11.2 | 14.5 | 19.9 | 12.5 |
| vs. Base | 0.0% | 0.0% | +6.4% | +42.4% | -10.4% | +16.0% | +59.2% | 0.0% |
| Code generation – Java | | | | | | | | |
| Av. len | 30.6 | 25.4 | 23.8 | 20.4 | 18.8 | 32.7 | 39.9 | 30.6 |
| vs. Base | 0.0% | -17.0% | -22.2% | -33.3% | -38.6% | +6.9% | +30.4% | 0.0% |
| Code generation – natural text | | | | | | | | |
| Av. len | 166.5 | 166.4 | 166.2 | 203.5 | 163.9 | 205.5 | 282.3 | 166.7 |
| vs. Base | 0.0% | -0.1% | -0.2% | +22.2% | -1.6% | +23.4% | +69.5% | 0.1% |
| Clone detection – Java | | | | | | | | |
| Av. len | 349.3 | 287.6 | 264.0 | 220.1 | 196.9 | 377.1 | 449.6 | 349.6 |
| vs. Base | 0.0% | -17.7% | -24.4% | -37.0% | -43.6% | +8.0% | +28.7% | +0.1% |
| Test pretraining data – Python | | | | | | | | |
| Av. len | 121.2 | 100.6 | 91.8 | 80.3 | 75.0 | 131.4 | 159.4 | 121.2 |
| vs. Base | 0.0% | -17.0% | -24.3% | -33.7% | -38.1% | 8.4% | 31.5% | 0.0% |
| Test pretraining data – Java | | | | | | | | |
| Av. len | 85.3 | 71.2 | 66.6 | 56.6 | 51.7 | 93.8 | 115.3 | 85.4 |
| vs. Base | 0.0% | -16.5% | -21.9% | -33.6% | -39.4% | 10.0% | 35.2% | 0.1% |

Table 4: Time required to generate predictions for the whole test set, in seconds. Numbers in brackets indicate relative speed-up versus Level 0. UnigramLM tokenization, 50k vocabulary.

| | Level 0 | Level 1 | Level 4 |
|---|---|---|---|
| Code translation (Python, 1693 examples) | 502 | 387 (77%) | 270 (54%) |
| Code generation (Java, 2000 examples) | 346 | 279 (80%) | 211 (60%) |

the loss of information. Note, however, that such inequality between subtokenizers does not devalue the results we report. First, the described clipping procedure corresponds to the conventionally used practical setting with the use of maximum available input information, and second, our results show

that longer (less compact) subtokenizations, most affected by clipping, generally perform *better* than shorter (more compact) subtokenizations which allow less information loss. Using the more "fair" setting may make the performance of more compact subtokenizations worse (and not affect less compact subtokenizations), amplifying performance differences further.

In this section, we conduct such a "fair" experiment for three representative downstream tasks and crop subtoken sequences produced by all subtokenizers so that they are all detokenized into a similar character sequence. In the example above, the 2K subtokenization will stay unchanged while the Level 4 subtokenization will be cropped into ['x =', 'sum ('] to achieve equality. The results are shown in Figure 7. Comparing the observations to the hypothesis given above, we observe that the performance of smaller vocabulary subtokenizations, less affected by cropping, stays similar (as expected) and that the performance of higher granularity subtokenizations, most affected by cropping, often reduces (Levels 0 and 1 in translation, Level 1 in summarization) or stays similar (Level 0 in summarization, Level 1 in clone detection), again as we expected. Surprisingly, in some cases (Level 4 in all tasks and Level 0 in clone detection), the performance of higher granularity subtokenizations improve slightly after cropping. We hypothesize that the reason may be that the last parts of sequences may be not important for correct prediction, e. g. `if __name__ == "__main__"` statements in Python translation examples may be auxiliary and not used during translation.

As for the conclusions emphasized in the main text, they hold for the cropped setting as well: (a) the marginal performance superiority of smaller vocabularies compared to the Level 0 50K vocabulary, is amplified (translation) or similar (summarization) as in the main text; (b) the relative performance of Level 0 and Level 4 is similar as in the main text; (c) the observation that the Level 1 subtokenization performs not worse than the Level 0 subtokenization stays same as in the main text.

Interestingly, in Java code summarization, Level 1 subtokenization performs slightly better than Level 0 subtokenization in the main text, while after cropping their performance is similar. We thus attribute the superiority of Level 1 in the main text to the less strong clipping than of Level 0 subtokenizations.

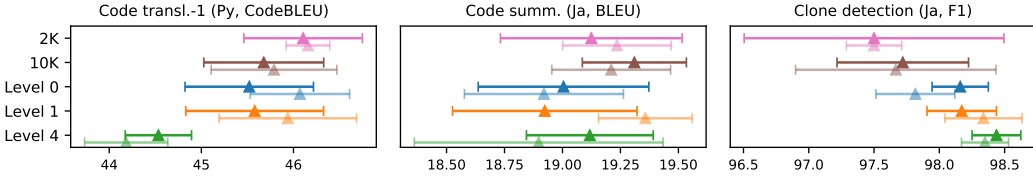

Figure 7: Additional experiments with subtoken sequences being cropped so that for each datapoint, the subtoken sequences produced by all subtokenizers are detokenized into a similar character sequence. Effectively this cropping means that the 2K subtokenizer's sequences stay unchanged while all others may be cropped, and the Level 4 subtokenizer's sequences are cropped the most. Blurred bars visualize original performance reported in the main paper, with "unfair" clipping used in practice. Both 2K and 10K subtokenizations use Level 0 preprocessing and all Level X subtokenizations use the 50K vocabulary, all experiments with UnigramLM.

**How much do input and output sequences intersect for various subtokenizations?** In attempt to better understand the effect of subtokenizations on downstream performance, we measure the Jaccard similarity $J(A, B) = |A \cap B|/|A \cup B|$ between the sets of input and output subtokens in sequence-to-sequence tasks. The intuition is that it should be easier for the model to predict correct subtokens if they are present in the input sequence. We only include textual subtokens in the considered sets (subtokens which do not include any programming language punctuation), since "copying" subtokens which include punctuation seems to be unrealistic (in text-to-code or code-to-text tasks, the text part does not include programming language punctuation at all, and in the translation task, punctuation of Python and Java are very dissimilar). We consider the setting with "fair" cropping described in the previous paragraph, however for the conventional setting the results are similar.

The results are presented in Table 5. The general trend that the higher the granularity, the lower the intersection rate, appears to be reasonable. For example, we observe the explainable monotonic

decrease in intersection for 2K–10K–50K vocabularies: the smaller the vocabulary, the smaller the granularity of identifiers/words subtokenization, the more chance that different parts of words will repeat. This correlates with our empirical observation that generally smaller vocabulary sutokenizations perform slightly better.

Interestingly, Level 1 subtokenization leads to a slightly higher intersection rate than Level 0. We explain it that punctuation combinations occupy a portion of vocabulary (3.4%) and thus reduce the effective vocabulary allocated for textual tokens, which will be slightly more often split into parts. *This effect shows that if one has some "length budget", it is better to be spent on splitting identifiers into subtokens rather than considering punctuation chars as separate tokens (they can be grouped).*

For further levels, two forces start competing: the first one that composite tokens occupy a part of the vocabulary and thus out-of-vocabulary identifiers are split into smaller pieces that are repeated relatively frequently, and the second one that composite tokens themselves are longer and are repeated rarely between input and output. The second force is absent for Level 1 considered above because Level 1 composite tokens are punctuation-only and do not include letters or digits. We observe that for Level 2, the first force outweighs the second one in translation and summarization (intersection rate is higher than for Levels 0 and 1) and underweights in generation (intersection rate lower than in Levels 0 and 1). Interestingly, this correlates with performance of Level 2 compared to Levels 0 and 1: it is on par with them in translation and summarization and lower in generation. Further on, at Levels 3 and 4 the intersection rate drops, with small increase of Level 4 compared to Level 3, explainable by the specifics of Level 3 construction focused on programming statements.

Finally, we compare the intersection rate of UnigramLM and BPE (both Level 0, 50K) and find that in summarization and generation it is slightly higher for UnigramLM than for BPE and in translation they are equal. This observation complements our findings that UnigramLM subtokenizations are better aligned with native subtokenizations than BPE subtokenizations and provides some intuition how it matters (similarity to native subtokenization presumably increases the number of repeated subtokens which should be easier to correctly predict).

Table 5: Average Jaccard similarities between the sets of input and output textual subtokens, for different subtokenizations. All subtokenizations expect the last one use the UnigramLM algorithm; Base ans Level 1–4 subtokenizations use vocabulary of 50K; 10K and 2K subtokenizations are based on Level 1 preprocessing.

|  | 2K | 10K | Level 0 | Level 1 | Level 2 | Level 3 | Level 4 | BPE |
|---|---|---|---|---|---|---|---|---|
| Code trans. (Py) | 14.61 | 13.91 | 13.68 | 13.81 | 14.38 | 6.65 | 7.79 | 13.67 |
| Code gen. (Ja) | 11.65 | 10.51 | 9.57 | 9.61 | 8.94 | 4.70 | 6.51 | 8.88 |
| Code sum. (Ja) | 10.25 | 7.26 | 5.99 | 6.03 | 6.33 | 2.84 | 5.80 | 5.67 |

**Subtoken frequences visualization.** In Figure 8, we visualize subtoken frequencies computed over 1/8 of the pretraining corpora for four subtokenizers: UnigramLM Level 0 50K, BPE Level 0 50K, UnigramLM Level 1 50K and UnigramLM Level 4 50K. Comparing UnigramLM and BPE, we find that UnigramLM has a slightly heavier tail which results in a greater number of subtokens having higher frequncies and thus better embeddings, the similar observation was also noticed in Bostrom & Durrett (2020). The frequency profiles of Level 0 and Level 1 subtokenizers are close. Level 4 vocabulary exhibits less contrast in frequencies than Levels 0 and 1: in the left range, Level 4 frequencies are lower, while in the right range, Level 4 frequencies are higher, which hypothetically could provide some advantage to Level 4 (but it does not, according to the reported performance results). We hypothesize that having high frequencies in the left range is important for high performance (more reliable "basic" subtokens) but high granularity subtokens reduce frequencies of smaller subtokens which negatively affects their embeddings.

# D NUMERICAL RESULTS

Table 6 presents the numerical results for figures in the main text and Appendix B.

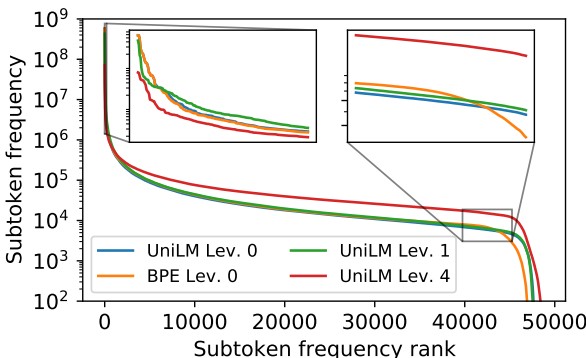

Figure 8: Subtoken frequencies over pretraining data.

Table 6: Numerical data for figures in the main text. CT1: Code Translation-1 (CodeBLEU), CT2: Code Translation 2 (Computational Accuracy), CS: Code Summarization (BLEU), CG: Code Generation (CodeBLEU), CD: Clone Detection (F1). Py – Python, Ja – Java.

| Subtokenizer | CT1 (Py) | CT1 (Ja) | CT2 (Py) | CT2 (Ja) | CS (Py) | CS (Ja) | CG (Ja) | CD (Ja) |
|---|---|---|---|---|---|---|---|---|
| UnigramLM 50K Level 0 | 46.1 | 48.2 | 65.3 | 57.1 | 19.7 | 18.9 | 38.2 | 97.8 |
| UnigramLM 50K Level 1 | 45.9 | 48.4 | 67.3 | 57.8 | 19.7 | 19.4 | 38.2 | 98.3 |
| UnigramLM 50K Level 2 | 45.9 | 48.0 | 67.0 | 56.8 | 19.5 | 19.3 | 37.3 | 98.2 |
| UnigramLM 50K Level 3 | 45.0 | 47.7 | 56.7 | 45.5 | 19.5 | 19.1 | 37.5 | 98.5 |
| UnigramLM 50K Level 4 | 44.2 | 46.7 | 54.3 | 43.7 | 19.5 | 18.9 | 36.7 | 98.3 |
| UnigramLM 10K Level 0 | 45.8 | 48.6 | 65.7 | 57.35 | 19.9 | 19.2 | 39.1 | 97.7 |
| UnigramLM 2K Level 0 | 46.2 | 48.0 | 66.1 | 56.2 | 19.8 | 19.2 | 39.1 | 97.5 |
| UnigramLM 50K Level 0 (Only Py) | 46.1 | 47.5 | 68.3 | 58.6 | 19.8 | 18.8 | 38.6 | 98.0 |
| BPE 50K Level 0 | 45.5 | 47.7 | 66.5 | 57.4 | 19.3 | 18.8 | 37.7 | 98.0 |
| BPE 50K Level 1 | 45.9 | 48.0 | 65.5 | 56.9 | 19.6 | 19.0 | 37.4 | 98.3 |
| BPE 10K Level 0 | 45.5 | 48.1 | 66.5 | 57.2 | 19.5 | 19.2 | 38.9 | 97.9 |