# OpenReview forum: "CodeBPE: Investigating Subtokenization Options for Large Language Model Pretraining on Source Code"
_ICLR.cc/2023/Conference — ICLR 2023 poster_

### Official Review · Reviewer_het6 · 2022-10-24

**Confidence:** 5
**Correctness:** 3
**Technical Novelty And Significance:** 2
**Empirical Novelty And Significance:** 4
**Recommendation:** 8

**Clarity, Quality, Novelty And Reproducibility:**

The paper is well written and easy to follow. The results are portrayed quite clearly. For the most part, the confidence intervals are rather wide and the results somewhat variable from one task to another, so that some of the findings do not readily permit any interpretation with statistical significance (e.g., Fig. 5). It may be worth emphasizing this in the discussion.

**Strength And Weaknesses:**

I reviewed a previous version of this paper that was similar to the current submission. As in my review then, I maintain that this is an interesting and novel paper. While it is perhaps more empirical in nature than most work published in ML venues, it provides many actionable insights and highlights the ways in which modeling source code is distinct from natural languages.

Given that the authors addressed my comments on the prior version, I have no significant weaknesses to note, other than, perhaps, that the model (PLBART) that all the results are based on is relatively small by current standards (two 6-layer Transformers). Given that this already required nearly 10K GPU hours due to the breadth of tasks and settings the model was trained for, I do not fault the authors.

**Summary Of The Paper:**

This work studies the impact of various sub-tokenization methods for neural language models trained on source code. Across a range of tasks and datasets, it demonstrates a number of actionable insights, such as that UnigramLM trained models perform better than BPE ones, and that certain types of tokenization (especially ones that fuse tokens across whitespace and newlines) tend to yield poorer downstream task performance in many settings, while others can be safely used to achieve both roughly comparable results while using fewer tokens to represent the same sized input.

**Summary Of The Review:**

This work presents actionable insights on a timely topic. It is rather empirical in nature, but comprehensive and well written. It would be good to publish it.

---

> ### Author Response · Authors · 2022-11-14
> **We thank the reviewer for the comments!**
>
> We thank the reviewer for the comments and efforts put into assessing our work!

---

### Official Review · Reviewer_SjVD · 2022-10-25

**Confidence:** 3
**Correctness:** 3
**Technical Novelty And Significance:** 2
**Empirical Novelty And Significance:** 3
**Recommendation:** 8

**Clarity, Quality, Novelty And Reproducibility:**

The paper is mostly well-written, but the presentation of the results could be a bit more precise in several places. For example, the abstract says that the proposed approach reduces average length by 17-40% without downstream performance drop. But 40% reduction is achieved with level 4, for which there is a significant performance drop. There are several such imprecise statements through out the paper, which makes the paper hard to read.

In figure 2, please use the same scale in the x-axis for the all the plots for easy comparison.

As an additional result, it would be useful to look at how does training and inference times change with the different tokenization schemes.



**Strength And Weaknesses:**

With the recent increase in pretrained large language models for code, investigating the effect of the different tokenization schemes is an important problem. This paper takes a step in that direction by thoroughly experimenting with different combinations of the design space and drawing useful insights.

Although the results have a huge variance and does not really show one choice of tokenization is fully superior over another, I commend the authors for the effort it took to do this thorough evaluation and believe, the results will be useful for the people in this community working on similar areas.

One concern I have is that all the evaluation was done with a single model, PLBART. And so, it is not clear if these insights drawn in this paper would apply to other kinds of models/sizes and other coding tasks.



**Summary Of The Paper:**

The paper implements and evaluates several tokenization schemes for pretraining large language models for coding tasks. The paper considers BPE vs UnigramLM algorithms, different vocabulary sizes, and different levels of composition allowed in tokens. Then they pre-trained PLBART-like models with these different tokenizations and compare results on several downstream tasks including code translation, code summary and code generation.


**Summary Of The Review:**

The experiment setup and evaluation is thorough and the paper presents previously understudied insights for the community, although it is not totally clear how the insights extend to other kinds of model architectures.

---

> ### Author Response · Authors · 2022-11-14
> **Answer to the reviewer's comments**
>
> We thank the reviewer for the comments and efforts put into assessing our work!
>
> * Evaluation only on PLBART
>
> We believe that even if exact relation in performance of one option compared to another could slightly vary for different models (for example higher levels of granularity may achieve similar performance to Level 0 for larger models), our work provides useful guidance which subtokenization options are more important to experiment with (granularity), and which — less (algorithm, vocabulary size). Repeating all experiments for another model would require much additional computational resources and have a significant negative environmental impact.
>
> * Numbers in abstract
>
> Thank you for pointing to this! Initially we meant that in some tasks, using Level 4 also performs similarly to Level 0, but we agree that in the current version of the text this statement is misleading, we reformulated it.

---

> > ### Comment · Reviewer_SjVD · 2022-11-18
> > **Reply to authors**
> >
> > Thanks for your response. Do you have any results/intuition on "how does training and inference times change with the different tokenization schemes?"?
> >
> > Also, my comment on writing is not just for the abstract. There are several such imprecise statements throughout the paper. And I also agree with the other reviewers that the writing can be more precise and intuitive.

---

> > > ### Author Response · Authors · 2022-11-23
> > > **Time measurement**
> > >
> > > Thank you for your reply!
> > >
> > > We measured the time needed to generate predictions for the test set, for Code translation (Java->Python) and Code Generation (Java). The measurement was conducted on a single Tesla V100 GPU, in the same session for all runs.
> > >
> > > | Evaluation time: seconds (speed-up vs L0) | Level 0 | Level 1   | Level 4   |
> > > |---------------------------------------------|---------|-----------|-----------|
> > > | Code translation (Python, 1693 examples)    | 502     | 387 (77%) | 270 (54%) |
> > > | Code generation (Java, 2000 examples)       | 346     | 279 (80%) | 211 (60%) |
> > >
> > > The speed-up is stronger than the length decrease, because of quadratic Transformer complexity. For example, for translation, the average length of Level 1 equals to 84% of the average length of Level 0 (see Table 3 in Appendix), while the inference time of Level 1 equals to 77% of the inference time of Level 0.
> > >
> > > We will include these statistics in Appendix.
> > >
> > > Regarding writing, we would appreciate it if you could give particular examples of the passages you found hard to understand.

---

### Official Review · Reviewer_RLnm · 2022-10-26

**Confidence:** 4
**Correctness:** 4
**Technical Novelty And Significance:** 2
**Empirical Novelty And Significance:** 2
**Recommendation:** 3

**Clarity, Quality, Novelty And Reproducibility:**

The presentation is clear though the presentation can be more insightful and inspiring. The paper has some empirical value.

**Strength And Weaknesses:**

(+) Studies an important under-studied problem.

(+) Experimental results are convincing

(-) It is not clear WordPiece is not included as several code-related model use it.

(-) While the numbers are interesting, the recommendation (using punctuation combination tokenization) appears somewhat obvious with hindsight – given the punctuation-heavy syntax of programming languages.


**Summary Of The Paper:**

Subtokenization is one of the unsung heroes of application of deep learning to code. The paper fills the gap by systematically investigating the results of several subtokenization approaches: BPE, UnigramLM, punctuation combination, native, and systematically reports statistics.

**Summary Of The Review:**

Subtokenization is an important aspect of models for code. The paper systematically explores the area and presents numbers which mostly confirm the the implicit assumptions of current models. The recommendation of punctuation combination approach is helpful information. Not including wordpiece is limiting.

---

> ### Author Response · Authors · 2022-11-14
> **Answers to the reviewer's comments**
>
> We thank the reviewer for the comments and efforts put into assessing our work!
>
> * Weakness 1: comparison to WordPiece
>
> Since the comparison of BPE and UnigramLM showed that the tokenization algorithm has a marginal influence on downstream performance, we decided not to consider other algorithms. We chose the comparison of BPE and UnigramLM following (Bostrom 2020), because these two algorithms represent two opposed paradigms: bottom-up (BPE) and top-down (UnigramLM) vocabulary construction (as described in Section 4). At the same time, WordPiece is basically a variant of BPE, with a slightly modified procedure for merging tokens.
>
> * Weakness 2: our recommendation on considering composite tokens
>
> We appreciate your opinion regarding the given recommendation. However, we would like to underline that the vast majority of existing pretrained LMs for source code do not include this technique in the pipeline, while it can result in quite a substantial speed-up in training, finetuning and inference. Since publicly released LMs are usually reused by other researchers and practitioners, we think it is important to show explicitly the advantage of allowing composite tokens and promote the release of models which include the proposed technique. Moreover, we propose various levels of composite tokens, while existing works, even if they consider composite tokens, only consider the highest granularity, which may be suboptimal, according to our experiments.

---

### Official Review · Reviewer_kR7n · 2022-11-02

**Confidence:** 4
**Correctness:** 3
**Technical Novelty And Significance:** 2
**Empirical Novelty And Significance:** 2
**Recommendation:** 5

**Clarity, Quality, Novelty And Reproducibility:**

The paper is hard to read and follow in some instances.
While it is known that vocabulary size and combination steps in BPE style algorithms impact the performance of a LM, it is hard to fully understand why it is the case for source code and exactly what gap are these general NL techniques addressing on source code.

Experiments on their own seem easy to reproduce.

**Strength And Weaknesses:**

Strengths
- this paper studies an interesting problem.
- authors consider code from multiple programming languages.

Weaknesses
- this paper is a little hard to follow at times, writing could be clear and so could the organization of sections in the paper.
- the paper primarily compares UnigramLM and BPE, with the authors claiming in the motivations that such tokenizers have demonstrated improvements in NL tasks. However, the authors fail to provide qualitative examples explaining why similar strategies achieve improvements on source code.
- while authors use the Sentencepiece vocabulary, they do not compare against Sentencepiece. I am not sure why the comparisons were restricted to UnigramLM and BPE alone.

**Summary Of The Paper:**

The authors of this paper propose various subtokenization strategies affecting input string length in a way so as to improve the efficiency of LLMs when trained on source code and when applied to downstream tasks such as code generation, code summarization and code clone detection.
Authors propose a strategy the restricts vocabulary size as well as compresses lengths of inputs without affecting performance via several combination strategies in the UnigramLM and BPE tokenization schemas.
The main contribution of this paper is to study the impact of tokenization for source code based applications.

**Summary Of The Review:**

While the problem is interesting, the presented analysis fails to provide insight into how proposed subtokenization strategies exactly help with source code.
Furthermore, I find it a little puzzling why comparisons against Wordpiece were omitted. Similarly while the point was to demonstrate the strengths of subword tokenizers, commentary on how they compare against other tokenization schema, i.e whitespace etc would be nice to see given that the vocabulary of source code is fairly limited.
On the whole while I see the experiments as thorough I am still unsure as to why these results need to be considered as novel.

---

> ### Author Response · Authors · 2022-11-14
> **Answers to the reviewer's comments**
>
> We thank the reviewer for the comments and efforts put into assessing our work!
>
> * Weakness 2 (why subtokenization achieves improvements on source code), comment regarding the comparison against other tokenization schema, i.e. whitespace-based tokenization, and the question what gap subtokenization addresses on source code:
>
> The question why subword-based tokenization performs substantially better than whitespace-based or char-based tokenization, has been widely studied in the literature, both for natural language (e. g. Sennrich 2016) and for source code (Karampatsis, 2020). With whitespace-based tokenization, rare words (in natural language) or identifiers (in source code) are replaced with an UNK token, and this hurts performance much. With character-based tokenization, sequences become too long so that it is hard for the model to learn patterns from the data and also such models are much slower than their counterparts. In both cases, the mentioned weaknesses are purely technical and same for natural language and source code, so one would not expect whitespace- or character-based models to perform on source code better than on natural language. Regarding the reviewers’ concern that “the vocabulary of source code is fairly limited”, we respectfully disagree, because source code contains a lot of programmer-defined variable names, function names and class names, the complexity of which is not restricted by any means (Chirkova, 2021). The presence of a large number of these identifiers is actually a reason why subtokenization is widely used for source code. We added the paragraph explaining these concepts in the introduction (highlighted in blue), thank you for the suggestion.
>
> Since subword-based tokenization is a standard part of all widely-used large pretrained models of natural language and source code (due to the reasons given above), we do not compare against whitespace-based and character-based tokenizations, but related experiments for source code can be found in (Karampatsis, 2020).
>
> * Weakness 3 (comparison to SentencePiece) and the comment regarding comparison against WordPiece
>
> SentencePiece is not a subtokenization algorithm but rather a library implementing both BPE and UnigramLM algorithms (please see https://arxiv.org/pdf/1808.06226.pdf for details). Regarding comparison to other tokenization algorithms, e. g. WordPiece, we did not consider them, because from the comparison of BPE and UnigramLM we saw that the tokenization algorithm has a marginal influence on downstream performance. We chose the comparison of BPE and UnigramLM following (Bostrom 2020), because these two algorithms represent two opposed paradigms: bottom-up (BPE) and top-down (UnigramLM) vocabulary construction (as described in Section 4). At the same time, WordPiece is basically a variant of BPE, with a slightly modified procedure for merging tokens.
>
> * Effect of vocabulary size for source code vs natural language
>
> As explained above, similarly to a wide variety of words in natural language, source code contains a lot of complex identifiers (variable names, function names etc). This is why subword-based tokenization is needed in source code processing and why decreasing the vocabulary size helps with composing those identifiers from smaller pieces, in the same way as it works in NLP (Ding, 2019).
>
> * how proposed tokenization exactly helps with source code
>
> From other reviewer’s comments, we would suggest that this refers to subword-based tokenization in general, advantages of which are discussed above and in Related work. At the same time, if this comment refers to the proposed punctuation merging approach, we provide qualitative analysis in Section 3 and Appendix B.
>
> * why these results need to be considered novel
>
> As we underline in Introduction and in Conclusion, though subtokenization is widely used for source code (Feng 2020, Kanade 2020, Ahmad 2021, inter alia), existing works do not provide experimental analysis on how they selected subtokenization options (allowed complexity of tokens, tokenization algorithm, vocabulary size etc). Our work is first to provide such analysis for the field, underlying which options have higher influence and which —- lower, and providing practical recommendations. At the same time, such recommendations cannot be directly borrowed from NLP, because source code has a much higher percentage of punctuation and of widely used code idioms, and the variety of identifiers in source code is much higher than of words in natural language.

---

> > ### Comment · Reviewer_kR7n · 2022-11-18
> > **Not entirely convinced with the authors responses**
> >
> > Thank you to the authors for clarifying.
> > It is true that the advantages are likely previously documented. However, that doesnt justify the lack of any qualitative examples to reinforce the same hypothesis as seen from the authors' own experiments. The authors do provide some support now, but if we go by what they has been seen in previous work the question on novelty lingers.
> > I acknowledge my typo on Sentencepiece. I intended to ask about WordPiece and not the Sentencepiece repo. However, that said by saying WordPiece is a variant of BPE and dismissing the comparison always leaves room for doubt. Both models differ in the way they combine subtokens, one will always wonder if the gains that WordPiece saw over BPE would translate to source code. So I disagree in dismissing it as a baseline.
> >
> > I am still not convinced how exactly is this work novel. Yes, we all know subword tokenization has several advantages in translation problems, with source code understanding and generation (like the authors themselves keep referring to) and with sequence tagging problems.
> > I do agree with an other reviewer who this this is a marginal contribution but not complete to warrant acceptance by itself.

---

> > > ### Author Response · Authors · 2022-11-23
> > > **Additional clarifications**
> > >
> > > Thank you for your comments!
> > >
> > > \> we all know subword tokenization has several advantages in translation problems, with source code understanding and generation. I do agree with an other reviewer who this this is a marginal contribution
> > >
> > > It seems that you misunderstood the intention of our explanation about subtokenization vs word level vs char level.  We do not claim in the paper that we propose using subtokenization for source code, this was proposed in (Karampatsis, 2020) and is now widely used in the source code processing community. Our work makes the next step and investigates different dimensions of subtokenization for code. For all considered dimensions, we do provide qualitative analysis and examples, as much as we found possible.
> > >
> > > Our explanation (and new paragraph in intro) was intended to give the broader background on the advantages of using subtokenization in general, which we assumed in our initial text the reader is already familiar with, because subtokenization is used in many deep learning models nowadays.
> > >
> > > Regarding WordPiece
> > >
> > > We appreciate your opinion. Unfortunately, we are unable to train a new PLBART with WordPiece tokenization now, due to technical reasons.

---

### Decision · Program_Chairs · 2023-01-20

**Decision:**

Accept: poster

**Justification For Why Not Higher Score:**

While the ideas discussed in this paper are broadly applicable, it relates only to a small subset of the community. In additional, due to (presumed) constraints, the authors do not compare how tokenization affects really large language models, which could have made this work interesting to a wider audience.

**Justification For Why Not Lower Score:**

This work is technically sound and exposes an under-explored issue that affects most transformer-based models of code. If this were to be rejected, the visibility of this work would be diminished. This would be unfortunate as transformer-based models of code (e.g. LLMs) are of interest to many in the community.

**Metareview: Summary, Strengths And Weaknesses:**

This work explores how different tokenization strategies affect transformer-based models of source code. It shows that the tokenization method has an impact on model performance for a variety of code-related tasks.

## Strengths
- Exposes an interesting problem in source code-related language models that has been overlooked since these methods were originally developed for natural language
- Shows that some tokenization methods can effectively reduce sequence size, allowing for generating longer sequences or improved computational speed.
- Provides actionable insights to anyone interested in training code-related LLMs and other transformer-based models.

## Weaknesses
- It is unclear how these observations are affected through model scale. For example, it's unclear if models with 100s of billions of parameters would have similar performance characteristics with respect to subtokenization. (Yet, sequence length will still be reduced)
-  A larger variety of subtokenization methods could have been explored.
- The presentation can be improved at some places (see reviewer comments)

## Other
A minor comment:
I would encourage the authors to include the SentencePiece training args in the appendix for the various settings they study.

**Note From Pc:**

if the above contains the word "oral" or "spotlight" please see: "oral" presentation means -> notable-top-5% and "spotlight" means -> notable-top-25%. As stated in our emails, we are disassociating presentation type from AC recommendations